Viscoelastic characteristics of the canine cranial cruciate ligament complex at slow strain rates

http://orcid.org/0000-0003-4887-9635 Readioff Rosti 1 r.readioff@leeds.ac.uk
Geraghty Brendan 2
Elsheikh Ahmed 1 3 4
http://orcid.org/0000-0002-5244-6042 Comerford Eithne 2 5 eithne.comerford@liverpool.ac.uk
1 School of Engineering, University of Liverpool , Liverpool , UK
2 Institute of Life Course and Medical Sciences, University of Liverpool , Liverpool , UK
3 Beijing Advanced Innovation Center for Biomedical Engineering, Beihang University , Beijing , China
4 UCL Institute of Ophthalmology, NIHR Moorfields BRC , London , UK
5 School of Veterinary Science, University of Liverpool , Neston , UK
Nganvongpanit Korakot
Electronic publication date: 2020 Dec 22
Publication date: 2020
Volume: 8
Electronic Location ID: e10635
Received 2020 Sep 18; Accepted 2020 Dec 2
Copyright: © 2020 Readioff et al.
Copyright year: 2020
Copyright holder: Readioff et al.
License: This is an open access article distributed under the terms of the Creative Commons Attribution License, which permits unrestricted use, distribution, reproduction and adaptation in any medium and for any purpose provided that it is properly attributed. For attribution, the original author(s), title, publication source (PeerJ) and either DOI or URL of the article must be cited.
License URL: https://creativecommons.org/licenses/by/4.0/

Keywords: Ligaments, Canine knee, Stifle joint, Cruciate ligaments, Viscoelastic, Strain rate, Hysteresis, Recovery

Funding: School of Engineering at the University of Liverpool (Rosti Readioff) Wellcome Trust Institutional Strategic Support Fund WT 204822/Z/16/Z (Eithne Comerford) National Institute for Health Research (NIHR) Biomedical Research Centre based at Moorfields Eye Hospital NHS Foundation Trust UCL Institute of Ophthalmology (Ahmed Elsheikh) This work was supported by the School of Engineering at the University of Liverpool (Rosti Readioff), the Wellcome Trust Institutional Strategic Support Fund (WT 204822/Z/16/Z) (Eithne Comerford), and by the National Institute for Health Research (NIHR) Biomedical Research Centre based at Moorfields Eye Hospital NHS Foundation Trust and UCL Institute of Ophthalmology (Ahmed Elsheikh). The funders had no role in study design, data collection and analysis, decision to publish, or preparation of the manuscript.

==============================
Ligaments including the cruciate ligaments support and transfer loads between bones applied to the knee joint organ. The functions of these ligaments can get compromised due to changes to their viscoelastic material properties. Currently there are discrepancies in the literature on the viscoelastic characteristics of knee ligaments which are thought to be due to tissue variability and different testing protocols.

The aim of this study was to characterise the viscoelastic properties of healthy cranial cruciate ligaments (CCLs), from the canine knee (stifle) joint, with a focus on the toe region of the stress-strain properties where any alterations in the extracellular matrix which would affect viscoelastic properties would be seen. Six paired CCLs, from skeletally mature and disease-free Staffordshire bull terrier stifle joints were retrieved as a femur-CCL-tibia complex and mechanically tested under uniaxial cyclic loading up to 10 N at three strain rates, namely 0.1%, 1% and 10%/min, to assess the viscoelastic property of strain rate dependency. The effect of strain history was also investigated by subjecting contralateral CCLs to an ascending (0.1%, 1% and 10%/min) or descending (10%, 1% and 0.1%/min) strain rate protocol. The differences between strain rates were not statistically significant. However, hysteresis and recovery of ligament lengths showed some dependency on strain rate. Only hysteresis was affected by the test protocol and lower strain rates resulted in higher hysteresis and lower recovery. These findings could be explained by the slow process of uncrimping of collagen fibres and the contribution of proteoglycans in the ligament extracellular matrix to intra-fibrillar gliding, which results in more tissue elongations and higher energy dissipation. This study further expands our understanding of canine CCL behaviour, providing data for material models of femur-CCL-tibia complexes, and demonstrating the challenges for engineering complex biomaterials such as knee joint ligaments.

Introduction

Ligaments play a major role in stifle (knee) joint stability (Budras, 2007; Levangie & Norkin, 2005), with part of the primary support being provided by the cranial cruciate ligament (CCL) (Carpenter & Cooper, 2000; Slatter, 2002). The CCL is the most commonly ruptured canine stifle joint ligament (CCLR) (Arnoczky, 1988; Gianotti et al., 2009) following acute injury or chronic disease, which can lead to destabilisation of surrounding structures and the subsequent development of osteoarthritis (Arnoczky, 1988; Bennett et al., 1988; Brooks, 2002; Comerford et al., 2006). There is a large economic cost associated with managing canine CCLR, for example in the United States alone the economic cost was estimated to be at least one billion dollars in 2003 (Wilke et al., 2005). Both human and canine CCL failure significantly increases the incidence of age-associated joint degeneration (Lee et al., 2014; Liu et al., 2003; Peters et al., 2018) and so understanding the tissue’s fundamental material properties can assist with the prediction and effective management of ligament injuries.

The phenomenon of viscoelastic characteristics including strain rate dependency, hysteresis, creep and stress relaxation has been observed consistently in soft biological tissues such as the sclera (Elsheikh et al., 2010; Geraghty et al., 2020), cornea (Elsheikh, Kassem & Jones, 2011; Kazaili, Geraghty & Akhtar, 2019) and tendon (Robinson et al., 2004; Zuskov et al., 2020). Similar to the other biological tissues, ligaments inherit viscoelastic characteristics meaning they exhibiting both elastic and viscous behaviour, hence they are history-and time-dependent (Bonner et al., 2015; Fung, 1993; Ristaniemi et al., 2018). The initial part of the non-linear load-deformation behaviour in ligaments is the toe region where the wavy collagen fibres become taut and straighten as load is applied, hence the crimp is removed (Fratzl et al., 1998). In this zone, there is a relatively large deformation of the tissue with little increase in load and this permits initial joint deformations with minimal tissue resistance (Amis, 2004; Dale & Baer, 1974; Fratzl et al., 1998; Wingfield et al., 2000) (Fig. 1).

Figure 1 A typical stress-strain curve of a knee joint ligament loaded to failure, illustrating the three major regions of the curve.

R1: toe region where the ligament fibres tighten and crimp is removed. R2: elastic region. R3: plastic region. The graph is based on previous literature including Amis (2004) and Wingfield et al. (2000).

Several studies tested for material characteristics of knee joint ligaments at traumatic loading rates and to failure loads (Crisco, Moore & McGovern, 2002; Crowninshield & Pope, 1976; Lydon et al., 1995). The loading rate is reported to be directly proportional to the tension which develops in ligaments (Pioletti, Rakotomanana & Leyvraz, 1999; Woo et al., 1990). This characteristic was also evident in a study investigating lower strain rates between approximately 2% and 54%/min representing physiological conditions other than trauma (Haut & Little, 1969). The study reported no change in the overall shape of the stress-strain curve, however, rapid change in the tangent modulus was found with the slow strain rates (between 1.7% and 10.8 %/min) and the change became progressively smaller with higher strain rates (above 10.8%/min). Similarly, it was reported that strain rate dependency decreases with the increase of deformation rate (Bonner et al., 2015; Crisco, Moore & McGovern, 2002). The stress-strain behaviour in the toe region (6% strain) showed strain rate dependency in canine CCL (Haut & Little, 1969). However, a study on rabbit medial collateral ligament complexes at varying strain rates (between 0.66% and 9300%/min) showed that the ligaments were only minimally strain rate dependent (Woo et al., 1990; Woo, Gomez & Akeson, 1981). The small effect of strain rate stiffening could be because the studies combined stress-strain characteristics at the toe region with the elastic region (Haut & Little, 1969; Ristaniemi et al., 2018).

During high strain rates, hysteresis (energy dissipated) in the ligament may protect the tissue from injury (Bonifasi-Lista et al., 2005). However, there are contradicting findings about hysteresis in soft biological tissues in relation to strain rates. Initially, hysteresis was believed to be weakly dependent on strain rates (Fung, 1993). In contrary, a study on the viscoelastic tensile response of bovine cornea showed an increase in hysteresis with decreasing strain rates (Boyce et al., 2007). It is suggested that Fung’s belief in this phenomenon was based on a small number of experiments on rabbit papillary muscle using only three different strain rates (Haslach, 2005). Hence, Fung’s findings only approximately support the independence of hysteresis from strain rates.

Therefore, there is a lack of understanding on the strain rate dependency and hysteresis of canine CCLs. Current information is limited to no clear methodological investigations on the strain rate dependency and hysteresis of the CCLs at the toe region (the initial part of non-linear load-deformation behaviour) where collagen fibres tighten and uncrimps with applied load, and importantly, any alterations in the extracellular matrix will be observed (Comerford et al., 2014; Lujan et al., 2009). Therefore, the purpose of this study was to characterise the viscoelastic properties of healthy canine CCLs as a femur-CCL-tibia complex, with a focus on the toe region of the stress-strain properties. This quantification is important when comparing the mechanical characteristics of the CCL and when developing synthetic, auto and allo-grafts to be used in future therapies for ligament replacement.

Materials and Methods

CCL storage and preparation

Cadaveric disease-free canine stifle joint pairs from skeletally mature Staffordshire bull terriers (n = 6 pairs) euthanatized for reasons other than musculoskeletal injury were obtained with full ethical permission from the Veterinary Research Ethics Committee ((VREC65), Institute of Veterinary Science, University of Liverpool). Inclusion criteria for cadaveric samples were a bodyweight >15 kg and age between 1.5 and 5 years old. The entire stifle joints were frozen at −20 °C until required and defrosted at room temperature prior to removing the CCLs as a femur-CCL-tibia complex (Readioff, 2017; Readioff et al., 2020). In order to harvest the femur-CCL-tibia complex, initially the stifle joints were dissected. Subsequently, approximately 10 mm of the femoral and tibial bones were left connected to the CCLs which allowed for the measurement of end-to-end ligament deformation as well as helping to facilitate the clamping of the specimen (Figs. 2 and 3).

Figure 2 The extracted cranial cruciate ligaments (CCL) consisted of approximately 10 mm of the femoral and tibial bones forming femur-CCL-tibia complex.

Figure 3 The uniaxial experimental test setup.

(A) The design of the custom-made clamps that was used for (B) manufacturing the parts of the experimental testing rig. The custom-built clamps included a (i) cylindrical Perspex tank, (ii) lower and (iii) top grips, (iv) duck-tail sandwich clamps and (v) screws. (C) An example of a cranial cruciate ligament (CCL) clamped and placed in the Perspex tank.

The extracted femur-CCL-tibia complexes were maintained in a moistened state in paper towels soaked with phosphate buffered saline (PBS; Sigma, Poole, UK) and frozen at −80 °C until they were required for testing (Woo et al., 1986). Prior to testing, the samples were thawed at room temperature and two 1.1 mm arthrodesis wires (Veterinary Instrumentation, Sheffield, UK) were drilled through the tibial and femoral bone ends (Fig. 2). These pins were placed to provide extra grip as well as to replicate the ligament’s slight proximal-to-distal outward spiral when secured using custom-built steel clamps (Arnoczky, 1983; Arnoczky & Marshall, 1977). The ducktail clamps were designed to provide a secure grip as well as ensuring that the CCLs were free and unobstructed throughout the experiment (Fig. 3). The clamped samples were then mounted on a mechanical testing machine.

CCL length

A modified version of a previously described method was used to determine the average length of CCL from the craniomedial and caudolateral portions of a ligament (Comerford et al., 2005; Vasseur et al., 1991). In this study, measurements between the insertion and origin of the CCLs at the cranial and caudal planes, as well as the lateral and medial planes were taken using a Vernier callipers (D00352, Duratool, Taiwan, China) accurate to ±10 µm. The mean values of these four length measurements were recorded to give an accurate record of the length of the CCL before deformation (Fig. S1).

CCL cross-sectional area

The method by Goodship and Birch was used to measure cross-sectional area (CSA) of the CCLs (Goodship & Birch, 2005). In brief, alginate dental impression paste (UnoDent, UnoDent Ltd., UK) was used to make a mould around the CCL. Once set, the mould was removed from the CCL and was used to create replicas of the CCLs. The replicas were cut into two in the middle and the surface of the replicas showing middle CSA were estimated using ImageJ (a public domain Java image processing program) (Fig. S2).

Mechanical testing

An Instron 3366 materials testing machine (Instron, Norwood, MA, USA) fitted with a 10 N load cell (Instron 2530-428 with ±0.025 N accuracy) was used to perform tensile tests. Initially, a preload of 0.1 N was applied to remove laxity within the CCL (Provenzano et al., 2002). Application of the preload was then followed by preconditioning the CCLs to ensure that they were in a steady state and would produce comparable and reproducible load-elongation curves (Butler, Noyes & Grood, 1978; Fung, 1993; Savelberg et al., 1993). Preconditioning involved performing ten loading-unloading cycles up to a maximum load of 10 N at 10 %/min strain rate (Ebrahimi et al., 2019; Woo et al., 1991). Subsequently the CCL was subjected to cyclic tensile loading-unloading tests investigating stress-strain behaviour of the ligament at the toe region through the application of 10 N load at sequential slow strain rates of 0.1%, 1% and 10%/min. Each strain rate consisted of three loading-unloading cycles which allowed for reproducible results. Between each two cycles, including the preconditioning procedure, a period of 6 min recovery time was given (Ebrahimi et al., 2019; Viidik, 1968). From the paired stifle joints, the left CCLs were exposed to an ascending strain rate test in which the rate of strain was increased from 0.1% to 1% and to 10%/min and the right CCLs were exposed to a descending strain rate in which the CCL was tested under decreasing strain rates from 10% to 1% and to 0.1 %/min (Pioletti, Rakotomanana & Leyvraz, 1999; Pioletti & Rakotomanana, 2000). A slow speed was chosen to better observe tissue response to loading at the toe region of load-deformation curves. The reverse orders of strain rate tests (ascending and descending strain rates tests) were carried out to identify characteristics associated with strain history of the ligaments at the toe region.

Data analysis

Analyses on the collected load-deformation data were performed using Microsoft Excel spreadsheets (Microsoft Office 2010, US) and MATLAB (MATLAB R2020a) (code for the analysis and graphs can be found in the Supplementary Materials). Nominal stress and strain values were estimated following Eqs. (1) and (2) (Haut & Little, 1969; Woo, Gomez & Akeson, 1981) and from these stress and strain values, tangent modulus values were determined (Eq. 3). Numerical integrations (the trapezoidal rule) on the stress-strain curves were used to estimate the stored energy in the ligaments during loading and unloading tests (Eq. 4). The hysteresis was then calculated from the difference between the stored energy during loading and unloading tests (Elsheikh et al., 2008) (Eq. 5). In addition, ligament extension before and after recovery were studied to investigate strain history and strain rate dependencies as a result of applying loads at different strain rate orders (loading at ascending or descending rates).

(1) σ=FCSA

where σ is stress in MPa, F is applied load in N and CSA is cross-sectional area at the middle of the CCL in mm2.

(2) ε=ΔLL0

where ε is strain, ΔL is change in length in mm (ΔL=L0−L1), L0 is initial length and L1 is deformed length of the CCL in mm.

(3) Etan=δσδε

where Etan is tangent modulus in MPa.

(4) U=∑k=1N12×(σk−1+σk)×Δεk

where U is the stored energy in MPa, N is the resolution of the trapezoidal partition, and Δεk is the length of the kth subinterval (Δεk=εk−εk−1).

(5) Hysteresis=ULoading−UUnloading

where ULoading and ULoading represent the stored energy during loading and unloading of the ligaments in MPa.

Statistical analysis

CCL lengths measured at different planes were categorised into cranial, caudal, medial and lateral groups. Statistical tests were performed using one-way analysis of variance (ANOVA) followed by a Bonferroni post-hoc test for multiple comparisons.

A two-tailed t-test (two samples with unequal variance) was used to test for differences between results obtained from the ascending and descending strain rate tests. In addition, one-way ANOVA followed by a Bonferroni post-hoc test for multiple comparisons was performed to test dependencies of tensile responses of the ligaments on strain rates. Distribution of data was illustrated in boxplots and suspected outliers were defined as any value greater than or equal to 1.5 times the interquartile range (range between the first and third quartiles).

All statistical analyses were performed in Microsoft Office Excel and 95% confidence level (p < 0.05) was selected to define significance for all statistical tests.

Results

CCL samples

The CCL samples (n = 6 paired stifle joints) used to investigate mechanical properties of the ligament were of mixed gender (female = 3 and male = 3) and the bodyweight of the cadavers were in the range of 17–25.5 kg (20.68 ± 3.85 kg).

CCL length

The lengths of the CCLs at different planes were in the range of 7.88–23.16 mm and the measurements at different planes of the individual ligaments are reported in Table 1.

Table 1 The measured length of cranial cruciate ligaments (CCL) at different measurement planes (cranial, caudal, medial and lateral) for CCLs in paired canine stifle joints (n = 6 pairs).

No.	Cranial plane (mm)	Caudal plane (mm)	Medial plane (mm)	Lateral plane (mm)	Average ± SD (mm)	
	Right	Left	Right	Left	Right	Left	Right	Left	Right	Left	
1	13.51	14.54	7.88	8.16	11.76	14.10	11.00	12.31	11.04 ± 2.35	12.28 ± 2.91	
2	22.79	22.07	11.20	12.00	17.00	13.1	20.54	16.93	17.88 ± 5.05	16.03 ± 4.55	
3	22.14	22.22	10.21	9.78	17.5	20.36	20.83	19.71	17.67 ± 5.34	18.02 ± 5.59	
4	21.44	23.16	13.53	11.55	17.51	19.05	16.05	14.37	17.13 ± 3.31	17.033 ± 5.12	
5	17.88	18.58	10.37	13.02	13.94	15.12	16.51	16.82	14.68 ± 3.30	15.89 ± 2.38	
6	15.30	17.83	9.20	9.38	13.50	15.81	13.1	12.31	12.78 ± 2.57	13.83 ± 3.74	
Mean ± SD (mm)	19.29 ± 3.47	10.52 ± 1.79	15.73 ± 2.60	15.87 ± 3.34			
Coefficient of variation (%)	18.0	17.0	16.5	21.0			

The ANOVA test showed statistically significant results in measuring CCL length in different plane views. The analysis showed that length measurements recorded at different planes were statistically different except for comparisons between medial and lateral planes (Table S1).

CCL cross-sectional area

The cross-sectional areas of the CCLs were in the range of 11.09–23.62 mm2 (16.1 ± 5.1 mm2) and cross-sectional areas of individual ligaments are reported in Table 2.

Table 2 The cross-sectional areas of cranial cruciate ligaments (CCL) for CCLs in paired canine stifle joints (n = 6 pairs).

No.	Cross-sectional area (mm2)	
	Right CCL	Left CCL	
1	12.58	14.99	
2	14.39	14.41	
3	11.09	29.08	
4	15.48	13.98	
5	12.91	15.69	
6	14.93	23.62	
Mean ± SD (mm2)	16.10 ± 5.10	
Coefficient of variation (%)	31.7	

Mechanical properties

Stress-strain

The stress-strain curves at 0.1%, 1% and 10%/min strain rates conformed to the typical non-linear behaviour as expected in canine CCLs (Haut & Little, 1969) (Fig. 4A). The stress-strain curves illustrated an increase in stress with increasing strain, and similarly an increase in stiffness was observed with increasing strain rates (Fig. 4B; Fig. S6). Although there was a small increase in stress with increasing strain rates, the increase was not statistically significant.

Figure 4 The tensile characteristics of a canine cranial cruciate ligament (CCL) following ascending (Asc) and descending (Desc) protocols at 0.1%, 1% and 10%/min strain rates.

(A) A typical cyclic loading and unloading stress-strain curves and (B) tangent modulus-stress behaviour of the loading curves of a CCL at varying strain rates.

The stress responses of the ligaments during ascending test protocol, where the cyclic loading commenced with 0.1%/min then increased to 1%/min and finally to 10%/min, were similar to responses during descending test protocol (the reverse of ascending test protocol). The stress-strain curves show that there are minimal differences in stress values between the two test protocols below 3% strain, and these differences become more distinguishable above 3% strain (Figs. 5A–5C). The testing protocols only minimally affected the stress-strain characteristics and not statistically significant. There are notably different mechanical behaviours among the specimens, as indicated by the grey dots in Figs. 5A–5C, and not all specimens reached 5% strain (Figs. S3–S5).

Figure 5 Tensile behaviour of the canine cranial cruciate ligaments was investigated following ascending (red line) and descending (black line) protocols at varying strain rates.

The box plots show specimen variation and stress at 1%, 3% and 5% strain during loading at (A) 0.1%/min, (B) 1%/min and (C) 10 %/min strain rates, and tangent modulus at 0.1, 0.3 and 0.5 MPa stress during loading at (D) 0.1%/min, (E) 1%/min and (F) 10%/min strain rates. The outliers are indicated with a red plus sign.

Tangent modulus-stress

Tangent modulus (Et), indicating the stiffness behaviour of the CCLs, increased with increasing stress (Fig. 4B) in both ascending and descending testing protocols. Similar to the observations from the stress-strain curves, the tangent modulus-stress lines between the two testing protocols (ascending and descending) were only minimally different and not statistically significant. Although not statistically significant, the increase in tangent modulus with strain rates was notable at higher stress values (Figs. 5D–5F). For example, at 0.5 MPa stress, average tangent modulus values from the ascending test protocol were 26.62, 31.40 and 32.66 MPa during loading at 0.1%, 1% and 10%/min strain rates.

Hysteresis

The results of this study show that hysteresis or dissipated energy are statistically different between the ascending and descending testing protocols (p = 0.0043). The mean values for hysteresis at 0.1%, 1% and 10%/min strain rates were 0.0032 (cycle 13), 0.0020 (cycle 16) and 0.0016 (cycle 19) MPa in ascending and 0.0040 (cycle 19), 0.0042 (cycle 16) and 0.0037 (cycle 13) MPa in descending testing protocol (Fig. 6A). The dissipated energy decrease from the first preconditioning cycle to the tenth (last) cycle was 85% for both testing protocols. In addition, hysteresis decreased with increasing strain rates (Fig. 6B). This characteristic was statistically significant during the ascending testing protocol (p = 0.039 between 0.1% and 1%/min and p = 0.013 between 0.1% and 10%/min). The metadata of hysteresis can be found in (Table S2).

Figure 6 Dissipated energy of the canine cranial cruciate ligaments (CCL) during cyclic loading at varying strain rates.

(A) The decrease in mean dissipated energy with increasing cycles of loads. The first ten cycles represent dissipated energy during the precondition stage of the CCLs. From cycles 11–19 dissipated energy values are associated with CCLs during tensile tests at three different strain rates (0.1%, 1% and 10%/min with each test repeated three times). The ascending testing protocol (red line) resulted in a slightly lower dissipated energy compared to the descending (black line) testing protocol. (B) Variations in dissipated energy within the specimens. The median values (indicated by the horizontal line inside the boxes) show a decrease in dissipated energy (hysteresis) with increasing strain rates. However, this observation was only statistically significant (blue line and *) between tensile tests at strain rates of 0.1 and 1%/min, and 0.1 and 10%/min.

Recovery

Length of the CCLs before and after the recovery period between each cycle showed consistent values during preconditioning and these values were in the range of 0.095–0.078 mm during ascending and 0.124–0.095 mm during descending tests (Fig. 7A). Unlike hysteresis, statistical analysis showed that tissue recovery was not different during ascending and descending tests. However, tissue length recovery was strain rate dependent and statistical analysis showed differences in recovery between 0.1% and 1%/min (p = 0.018) and 0.1% and 10%/min (p = 0.001) (Fig. 7B).

Figure 7 Recovery of the canine cranial cruciate ligaments (CCL) during cyclic loading at varying strain rates.

(A) The average recovered length of the CCLs at different cycles. During the preconditioning cycles (the first 10 cycles) recovered lengths of the ligaments are similar. Cycles associated with mechanical tests (cycles 11–19) for both testing protocols (ascending in red and descending in black) showed an increase in recovery with increasing strain rates. (B) Variations in length recovery within the CCLs. The box plots show an increase in recovery with increasing strain rates. This characteristic was statistically significant between tensile tests at strain rates of 0.1 and 1%/min, and 0.1 and 10%/min as indicated by the blue asterisk (*) and line.

Discussion

The aim of this study was to gain a greater understanding of the viscoelastic behaviour of the canine CCLs as a femur-CCL-tibia complex at the toe region of the stress-strain curves in order to better mechanically detail the material for its use in developing future therapies. Therefore, we carried out an experimental study investigating the nonlinear viscoelastic properties of CCLs, namely strain rate dependency, hysteresis and recovery, from healthy canine stifle joints. The findings in this study are the first to report the slow strain rate dependency of the canine cruciate ligament across three orders of magnitude with ascending and descending test arrangements. A previous study showed that with high strain rates, the toe region of stress-strain curves appears at lower strain levels (Haut & Little, 1969), however in order to study the toe region in detail without being limited to the level of strain, slow strain rates (≤10%/min) were utilised during mechanical tests in the current study.

Similar to previous studies, the measured CCL length and cross-sectional area were used in the calculations for stress and strain values (Wingfield et al., 2000). The difference in the CCLs length in medial and lateral planes was smaller than other planes, and this could be due to the anatomical structure of the femur-CCL-tibia complex.

The non-linear stress-strain pattern for the CCLs is consistent to that previously reported in studies on biological tissues such as tendons and ligaments (Bonner et al., 2015; Crisco, Moore & McGovern, 2002; Haut & Little, 1969; Pioletti, Rakotomanana & Leyvraz, 1999; Pioletti & Rakotomanana, 2000). The stress-strain and tangent modulus-stress characteristics of the CCLs were similar during the ascending and descending testing protocols (Figs. 4 and 5). These findings are similar to a previous study on bovine anterior cruciate ligament-bone complex where specimens were loaded up to 300 N at seven different strain rates (6%, 60%, 300%, 600%, 1,200%, 1,800% and 2,400%/min) and then tested for strain rate order by reloading the ligaments at the 6% and 300%/min strain rates (Pioletti, Rakotomanana & Leyvraz, 1999). They found identical stress-strain behaviour for the initial and reloaded specimens suggesting no difference in changing test protocols via strain rate orders. Their study applied higher strain rates (6–2,400%/min) than those used in the current paper and they reloaded the tissue in an ascending strain rate order only. Pioletti, Rakotomanana & Leyvraz (1999) did not study tissue hysteresis or recovery, but they reported increases in linear tangent moduli with increasing strain rates (although not statistically significant) which is similar to the findings in the current study.

The mechanical response of human knee ligaments to loading depends on strain rate which is less pronounced at lower rates (Dorlot et al., 1980; Van Dommelen et al., 2005) and this was also observed in the current study. It is believed that during lower strain rates the collagen fibrils in patella tendons undergo significantly less recruitment (Clemmer et al., 2010) and this could potentially be similar in the case of the CCLs. At slow strain rates (≤10%/min) the collagen fibrils uncrimp with applied load and then show intra-fibrillar gliding (Bonner et al., 2015; Karunaratne, Li & Bull, 2018). However, at fast strain rates (≥300%/min) fibrils go from an unloaded state directly to intra-fibrillar gliding where the matrix bond between the collagen molecules are broken before the removal of collagen crimps (Bonner et al., 2015). This could mean that the extracellular matrix components such as proteoglycans, which is directly linked to the mechanics of ligaments during uncrimping of the collagen fibres, might not affect the mechanical response during loads at higher strain rates.

In our study, hysteresis, which represents the dissipation of energy within the tissue, has shown some dependencies on strain rates and decreases with increasing strain rates (Fig. 6B). This finding contradicts the conclusions from previous literature (Bonifasi-Lista et al., 2005; Woo, Gomez & Akeson, 1981) but agrees with a recent study on tendon fascicle mechanics (Rosario & Roberts, 2020) and a study on bovine cornea (Boyce et al., 2007) which found a decrease in strain energy storage with increased loading rate. The discrepancies in results might be due to following different testing protocols, in particular the rate of applied loads. In the current study, where slow strain rates of ≤10%/min were used, the tissue goes through more steps (uncrimping collagen fibres, intra-fibrillar gliding and then loading of collagen respectively with applied loads) during lower strain rates (Bonner et al., 2015), and this slow process results in more tissue elongations hence higher energy dissipation. Energy dissipation was highest during the first precondition cycle and this could be a result of tissue handling and loading history of the CCLs (Fig. 6). In addition, it is possible that the ligament was dissipating higher energy at the initial loading cycles (during preconditioning) because of microstructural reorganisation. Similar to previous literature, this study found a decrease in hysteresis during preconditioning cycles (Woo et al., 1986; Yahia & Drouin, 1990).

The CCLs showed higher length recovery during higher strain rates compared to the lower strain rates (Fig. 7). This behaviour could be a result of resilience in the fascicular level of the tissue. It has been reported that during slow strain rates, the resilience in the fascicular level is lowest (Rosario & Roberts, 2020) which could lead to higher changes in the microstructural organisation of the tissue. Hence, the tissue’s length might not fully recover within the same recovery time as during higher strain rate tests. However, it is important to note that although the higher strain rate might seem to result in a more recovered ligament length, it is possible that the tissue collagen fibres are still crimping back as a result of previous loading history or insufficient recovery time. Further study in this area especially at the microstructural level of knee ligaments is necessary to better understand the effects of loading rates on the organisations of the fibres and extracellular matrix.

There were several limitations to our study. Preparing specimens for mechanical tests as a whole unit (femur-CCL-tibia complex) might have introduced some limitations such as overlooking the complexity of the anatomical structure of the CCLs which consists of two fibre bundles (caudolateral (CLB) and craniomedial bands (CMB)) functioning independently from one another in stifle joint flexion and extension (Arnoczky & Marshall, 1977; Carpenter & Cooper, 2000). Independent functioning of the CLB and the CMB allows the fibre bundles to reach their maximum potential (Arnoczky & Marshall, 1977; Carpenter & Cooper, 2000; Tanegashima et al., 2019). However, it is important to note that these two fibre bundles are not structurally segregated within the tissue, thus allowing the ligament to function as a united structure (Heffron & Campbell, 1978). In addition, the approximation methods adopted to measure the cross-sectional area and length of the specimens might be considered as another limitation. Further investigation with a larger number of specimens might improve the reliability of the statistical analysis and provide a broader view on the effect of cadaveric demography (i.e., age, gender, bodyweight) on the mechanical properties and microstructural organisations of knee ligaments (Duval et al., 1999; Woo, Ohland & Weiss, 1990; Woo et al., 1990). In addition, investigating material properties of the ligaments using other quantitative measures such as stress relaxation or creep could further expand our knowledge on the viscoelasticity of the tissue (Amis, 1985).

Conclusions

The current study focused on the viscoelastic behaviour, such as strain rate dependency, hysteresis and recovery of canine CCLs at slow strain rates to better understand the tissue behaviour at the toe region where the constituents of the extracellular matrix makes a major contribution to ligament mechanics.

Our changing test protocols via strain rate orders only affected hysteresis which might be a result of the strain history of the tissue or high-level of biological variability across samples (Gardiner & Weiss, 2003; Harris et al., 2016). The stress-strain of the CCLs at the toe-region associated with the extracellular matrix of the ligaments was not strain rate dependent. However, hysteresis and recovery were strain rate dependent and this is likely due to changes in microstructural organisation of the ligaments during mechanical tests.

The result of our study indicates the need for further investigations on the viscoelastic behaviour of the canine CCLs when loaded with different orders of strain rates, with a focus on extracellular matrix and collagen fibre organisations.

Supplemental Information

Supplemental Information 1 Supplementary Materials.

Click here for additional data file.

We thank Mr. Lee Moore, Mr. Ben Jones and the staff at Veterinary Teaching Suite, School of Veterinary Science for their assistance during sample collection. We also thank Mr. John Curran at School of Engineering, University of Liverpool, for their assistance during manufacturing parts of the experimental setup.

Additional Information and Declarations

Competing Interests

Author Contributions

Animal Ethics

Data Availability

The authors declare that they have no competing interests.

Rosti Readioff conceived and designed the experiments, performed the experiments, analyzed the data, prepared figures and/or tables, authored or reviewed drafts of the paper, and approved the final draft.

Brendan Geraghty analyzed the data, authored or reviewed drafts of the paper, and approved the final draft.

Ahmed Elsheikh conceived and designed the experiments, authored or reviewed drafts of the paper, and approved the final draft.

Eithne Comerford conceived and designed the experiments, authored or reviewed drafts of the paper, and approved the final draft.

The following information was supplied relating to ethical approvals (i.e., approving body and any reference numbers):

Cadaveric canine stifle joints were obtained with full ethical permission from the Veterinary Research Ethics Committee ((VREC65), Institute of Veterinary Science, University of Liverpool, Liverpool, UK).

The following information was supplied regarding data availability:

Raw, analysed and statistical data are available at Zenodo:

Readioff, Rosti. (2020). Raw data for viscoelastic anterior cruciate ligament characteristics (Version v2.0) (Data set). Zenodo. DOI 10.5281/zenodo.4264834.

Codes used for the initial data analysis and plotting the graphs are available at Zenodo:

Rosti Readioff. (2020, November 9). RostiReadioff/RReadioff_Viscoelasticity_of_Knee_Ligaments: Codes for plotting and analysing VE ACL data (Version v2.0). Zenodo. DOI 10.5281/zenodo.4264881.

Additional figures and tables are available as a Supplemental File.

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
