# Peer review of "Viscoelastic characteristics of the canine cranial cruciate ligament complex at slow strain rates"

_PeerJ, doi:10.7717/peerj.10635_

## Round 0.1 · original submission · Major Revisions

Your study is very interesting, however, we need a revised version for improving your manuscript. Please, correct your manuscript following all comments from reviewers point by point.

Reviewer 1 ·

Basic reporting

The article is written with clear and unambiguous, professional English and technically correct text, conforming to professional standards of courtesy and expression.
The authors include sufficient introduction and background providing Relevant literature references that fit into the broader field of knowledge.
Professional article structure is used in this article. All figures are relevant to the content of article and have sufficient resolution with appropriate descriptions and labels. The article is self-contained with relevant results to hypotheses but
Self-contained with relevant results to hypotheses.
One important raw data should be added. As dissipated energy or hysteresis is an important finding in this research work, therefore it will be useful if the authors can provide the detailed calculation of dissipated energy in Materials & Method (Data analysis) for future readers to clearly understand your work.

Experimental design

The article is original primary research within Aims and Scope of the journal.
Research question is well defined which is relevant & meaningful. It is stated how research fills an identified knowledge gap. By using slow strain rates, the authors investigate quite valuable results that can fill the knowledge gap and discrepancies which have been recently occurred in this research field.
The investigations in this article have been conducted rigorously and performed to a high technical standard. The prevailing ethical standards in the filed such as a full ethical permission from the Veterinary Research Ethics Committee ((VREC65), Institute of Veterinary Science, University of Liverpool).
Methods in terns of Cranial cruciate ligament storage and preparation, Cranial cruciate ligament length, Cranial cruciate ligament cross-sectional area and mechanical testing are well described with sufficient information to be reproducible by another investigator except Data analysis. The detailed calculation and analysis of hysteresis or dissipated energy should be added as also mentioned in 1 Basic reporting.

Validity of the findings

The article contains findings that should have high impact and novelty in the relevant research field. The authors clearly described all meaningful replication encouraged where relational & benefit to literature is clearly stated. The authors provide the useful raw data in supplemental materials, including P-values obtained from Bonferroni post-hoc statistical analysis on ligament lengths at different planes, appropriate figure of specimen (The cranial cruciate ligament (CCL) from a right canine stifle joint) showing the different plane views, the middle cross-sectional area (CSA) of the cranial cruciate ligament (CCL) estimated by firstly creating an alginate paste and the stress-strain characteristics of individual canine cranial cruciate ligaments (CCL) during the ascending (Asc) and descending (Desc) tests at strain rate of 0.1, 1 and 10 %/min. However, the author missed to provide the metadata of dissipated energy or hysteresis of each stress-strain curve as this is important in the current research work and this data should add value to the literature.
Conclusions are appropriately stated, connected to the original research question investigated and limited to supporting results. Claims of a causative relationship is supported by a well-controlled experimental intervention.

Additional comments

This manuscript contains original results of strain rate dependence on the mechanical properties of the canine cranial cruciate ligament complex at slow strain rates (0.1, 1, 10 %/min). The authors claimed that the findings in this study are the first to report the slow strain rate dependency of the canine cruciate ligament across three orders of magnitude with ascending and descending test arrangements. The authors aware of their small number of specimens which have only 6 pairs stifle joints from skeletally mature Staffordshire bull terriers, however, their findings are quite valuable to future readers in this related field. For examples, the statistically significant effects of strain rate were observed on the dissipated energy and the recovery of ligament lengths of the testing protocols, this preliminary result may be useful for further studies, such as investigations on the viscoelastic behaviour of the canine CCLs when loaded with different orders of strain rates or investigations of microstructural organisation of the ligaments during mechanical tests in larger number of specimens etc. This quantification is important when comparing the mechanical characteristics of the CCL and when developing synthetic, auto and allo-grafts to be used in future therapies for ligament replacement. In order to improve the manuscript before publishing I have some additional comments for the authors to consider as in following points.
1. As the toe region of stress-strain curve is important in this work. It should be useful if the authors can provide an example of a stress-strain curve of typical viscoelastic material or if possible the chosen CCL specimen that can illustrate three major regions of the stress strain curve: 1) the toe or toe-in region, 2) the linear region and 3) the yield and failure region to help readers better understand your focus point.
2. From the result of Cranial cruciate ligament length, in paragraph; “The ANOVA test showed statistically significant results in measuring CCL length in different plane views. The analysis showed that length measurements recorded at different planes were statistically different except for comparisons between medial and lateral planes (Supplementary Material (Table 1)).”, please explain in more detail why the difference in the CCL lengths in medial and lateral planes is smaller than other planes.
3. From these sentences “Prior to testing, the samples were thawed at room temperature and two 1.1 mm arthrodesis wires (Veterinary Instrumentation, Sheffield, UK) were drilled through the tibial and femoral bone ends (Fig. 1). These pins were placed to provide extra grip as well as to replicate the ligament’s slight proximal-to-distal outward spiral when secured using custom built steel clamps (Arnoczky, 1983; Arnoczky and Marshall, 1977)“ in Materials and Method (Cranial cruciate ligament storage and preparation), it is quite difficult to follow why these two pins can provide extra grip as shown in Fig. 1. I suggest that the schematic diagram showing the way these pins placed should be useful for clearly understand this point.
4. From Fig. S3 to S5, the stress-strain characteristics of individual canine cranial cruciate ligaments (CCL) during the ascending (Asc) and descending (Desc) tests at 0.1 to 10%/min strain rates, respectively, the label in all graphs of ligament specimens No. 6 should be CCL6_Desc instead of S6_Desc.

Reviewer 2 ·

Basic reporting

1. In the introduction part from line 62 to line 64: "Similarly, ligaments inherit non-linear viscoelastic characteristics exhibiting both elastic and viscous behaviour, hence they are history- and time-dependent". Nonlinear viscoelasticity is complicated, The viscoelastic specimen can also show elastic & viscous behavior in its linear viscoelastic regime. More clarification is needed here.

Experimental design

1. How do the authors ensure each tested sample has the similar dimension between different specimens?

2. Is the increase in strain a step change or a continuous change? If the increase is a step increase, what is the interval between each step? Because difference in these intervals will allow different time for the sample to relax and lead to differences in test results.

Validity of the findings

1. How did the authors account for the change of cross-sectional area during the measurement to calculate stress? Or the authors are using engineering stress rather than true stress.

2. In Figure 3, the authors show the nonlinear stress-strain relation. Could the authors fit these nonlinear functions or plot the data in semi-log scale to see if the curves have any similarities in stress-strain behavior?

3. Following the above question, it would also be helpful to show the unloading part of the stress-strain curve to demonstrate the viscoelastic behavior.

4. Another more direct and quantitative way to measure viscoelasticity is to conduct a stress relaxation or creep measurement instead of just measuring the stress-strain curve with only changes in strain rate.

5. Most energy dissipation in Figure 5 takes place after the first precondition cycle and plateaus towards the 20th cycle with few changes. Could the authors elaborate more on this behavior?

---

## Round 0.2 · accepted · Accept

Congratulations on your accepted manuscript.

Reviewer 1 ·

Basic reporting

-

Experimental design

-

Validity of the findings

-

Additional comments

-

Reviewer 2 ·

Basic reporting

no comment

Experimental design

no comment

Validity of the findings

no comment

Additional comments

All my comments are addressed in the revised manuscript.